# Insights into the Transcriptional Regulation of Branching Hormonal Signaling Pathways Genes under Drought Stress in Arabidopsis

**DOI:** 10.3390/genes12020298

**Published:** 2021-02-20

**Authors:** Nkulu Kabange Rolly, Bong-Gyu Mun, Byung-Wook Yun

**Affiliations:** 1Laboratory of Plant Functional Genomics, School of Applied Biosciences, Kyungpook National University, Daegu 41566, Korea; rolly.kabange@gmail.com (N.K.R.); mun0301@naver.com (B.-G.M.); 2Department of Southern Area Crop Science, National Institute of Crop Science, RDA, Miryang 50424, Korea; 3National Laboratory of Seed Testing, National Seed Service, SENASEM, Ministry of Agriculture, Kinshasa 904KIN1, Democratic Republic of the Congo

**Keywords:** PIN-FORMED, more axillary branching, isopentenyltransferase, gibberellic acid, *AtbZIP62*, transcription factor, drought tolerance, *Arabidopsis*

## Abstract

A large number of hormonal biosynthetic or signaling pathways genes controlling shoot branching are widely known for their roles in regulating plant growth and development, operating in synergetic or antagonistic manner. However, their involvement in abiotic stress response mechanism remains unexplored. Initially, we performed an in silico analysis to identify potential transcription binding sites for the basic leucine zipper 62 transcription factor (bZIP62 TF) in the target branching related genes. The results revealed the presence of *cis-*regulatory elements specific to two bZIP TFs, *AtbZIP18* and *AtbZIP69*, rather than *AtbZIP62*. Interestingly, these bZIP TFs were previously proposed to be negatively regulated by the *AtbZIP62* TF under salinity in *Arabidopsis*. Therefore, we investigated the transcriptional regulation of more axillary branching (MAX, strigolactone), PIN-FORMED (PINs, auxin carriers), gibberellic acid (GA)-biosynthetic genes as well as isopentenyltransferase (IPT, cytokinin biosynthesis pathway) genes in response to drought stress in *Arabidopsis* Col-0 wild type. In addition, in the perspective of exploring the transcriptional interplay of the selected genes with the *AtbZIP62*, we measured their expression by qPCR in the *atbzip62* (lacking the *AtbZIP62* gene) background under the same conditions. Our findings revealed that the expression of *AtMAX2*, *AtMAX3*, and *AtMAX4* was differentially regulated by drought stress between the *atbzip62* and Col-0 wild type, but not *AtMAX1.* Similarly, the transcripts accumulation of *AtPIN3* and *AtPIN7* (known as auxin efflux carriers), and that of the *AtAXR1* showed similar regulation patterns in *atbzip62*. However, *AtPIN1* expression was downregulated in Col-0, but no change was observed in *atbzip62*. Furthermore, *AtIPT5* and *AtIPT7* exhibited a differential transcripts accumulation pattern in *atbzip62* and Col-0 wild type (WT). In the same way, the expression of the GA biosynthetic genes *AtGA2ox1* and *AtGA20ox2,* and that of *AtRGA1* were differentially regulated in *atbzip62* compared to the Col-0. Meanwhile, *AtGA2ox1* showed a similar expression pattern with Col-0. Therefore, all results suggest PIN, MAX, IPT, and GA-biosynthetic genes, which are differentially regulated by *AtbZIP62* transcription factor, as emerging candidate genes that could be involved in drought stress response mechanism in *Arabidopsis*.

## 1. Introduction

Due to their sessile nature, plants are often subjected to various abiotic stresses induced by enviromental stimuli. Abiotic stresses cause major loss to crop yield [1,2,3], and drought stress is considered as one of the major threats to the life and productivity of plants [4]. Drought can impair the growth of the plant in various ways, leading to changes in metabolic functions. One among them is the deteroriation of the photosynthetic pigments resulting in a reduced light harvesting capacity, which ultimately results in the reduction of plants biomass [4]. However, like other abiotic stresses, the key impact of drought stress is the genereation of highly reactive and toxic molecules known as reactive oxygen species (ROS) [5,6], which have the ability to induce oxidative stress. The increased production of ROS during environmental stresses may cause oxidative damage, leading to peroxidation of lipids, oxidation of protiens, inhibition of enzymes, damage to nucleic acids, and induction of programmed cell death (PCD) that culminates in cell death [7,8,9,10].

To survive, plants have developed sophisticated mechanisms, including the activation of antioxidant (enzymatic and non-enzymatic) systems [11,12]. Both systems are assumed crucial to maintain at a controlled level the accumulation of ROS, while tending to maintain a balanced reduction-oxidation state within the cell [13]. Under the same conditions, a transcriptional reprogramming within the cell occurs, which includes the activation or suppression of stress responsive genes, coupled with an active interaction between genes or between proteins and DNA [14,15,16].

*Arabidopsis* has been recognized as the model plant species for dicots [17,18,19], and has served as an ideal plant species for plant biosciences research in the last two decades. This Brassicaceae offers a wide range of opportunities to study molecular functions of genes, in part due to its relatively small genome size and its short life cycle, coupled with the available T-DNA insertion lines generated by the Arabidopsis Biological Research Center (ABRC).

Transcription factors (TFs) are regulators of the expression of genes in biological systems. Intrinsic to their mode of action is their ability to bind to *cis*-regulatory elements found in the promoter region of genes [20], operating either alone or in complex with other molecules to activate or repress the recruitment of the basal transcriptional machinery to specific genes [21], thereby determining when and where the target genes are transcribed, how many proteins are synthesized, and what the phenotype looks like. Interactions between proteins and DNA are fundamental to nearly all biological processes of all biological systems [22]. Basic leucine zipper (bZIP) proteins [23] are transcription factors (TFs) involved in diverse developmental processes, including plant growth, flowers development, seeds maturation, and signaling during abiotic and biotic stresses [24].

In *Arabidopsis thaliana* (*Arabidopsis* hereafter) genome, about 75 distinct members of the bZIP family have been reported [23], including the *AtbZIP62* [24]. *AtbZIP62* belongs to the group I of bZIP TFs superfamily having the G-box binding factor 1 (GBF1)-like domain. The GBFs contain an N-terminal proline-rich domain in addition to the bZIP domain. GBFs have been also reported by Ábrahám et al. [25] to be involved in the developmental and physiological processes in response to various stimuli, such as light or hormones.

For several years, phytohormones have been shown to play fundamental and diverse roles in the metabolism of plants, including seed dormancy and germination, plant growth and development, flowering and organogenesis, seed formation and maturation, fruit ripening, and signaling during abiotic or biotic stress occurrence [26]. However, their possible involvement in the adaptive response mechanism under abiotic stress remained unexplored for several decades.

Therefore, this study aimed at investigating the transcriptional regulation of key hormonal biosynthetic or signaling pathways genes, previously known for their roles in the regulation of shoot branching in plants, in response to drought stress. In this perspective, we monitored by qPCR the transcripts accumulation of auxin carriers (PIN-FORMED protein) encoding genes, strigolactone biosynthetic genes (more axillary branching, MAX), gibberellic acid (GA) biosynthetic genes, and cytokinin (CK) biosynthetic genes (isopentenyltransferase, IPT) in *Arabidopsis* Col-0 wild type under drought stress conditions. In addition, in order to explore the transcriptional interplay of the target genes with the *AtbZIP62* TF, their expression levels were analyzed in the *atbzip62* knockout plants (lacking the *AtbZIP62* gene recently suggested to be involved in the adaptive response towards drought [27] and salinity [28] tolerance) compared to the Col-0 wild type (WT). Moreover, the phenotype of *atbzip62* plants was examined under normal growth conditions.

## 2. Results

### 2.1. AtbZIP62 TF Could Be Involved in the Control of Bud Outgrowth in Plants

Initially, we were interested to see the phenotype of the *atbzip62* knockout plants under normal growth conditions. Therefore, we measured the growth related parameters and the productivity of the *atbzip62* compared to that of the Col-0 wild type. Interestingly, our data show that *atbzip62* plants had a high branching phenotype under normal growth conditions. In essence, we recorded a significant increase in the number of tillers per plant (27.5%) compared to Col-0 WT (Figure 1A,H) and the number of secondary branches per plant (activated bud outgrowth) (12.6%) (Figure 1B). Consequently, more siliques were formed per branch (49.3%) (Figure 1C) and more siliques were produced per plants (41.4%) (Figure 1D), which profoundly resulted in more seeds per silique (3.7%) (Figure 1E) and more seeds per plant (49.6%) (Figure 1F). In addition, we recorded an increase in seeds weight by 17.4% compared to Col-0 WT (Figure 1G). We also observed that the *atbzip62* had taller plants compared to Col-0 WT. For this reason, we were interested to visualise the difference in the plant height between the *atbzip62* and the *atgsnor1-3* (known to have a stunt and high branching phenotype). As shown in the panels I and J of the Figure 1, *atbzip62* and *atgsnor1-3* plants exhibited distinctive growth habits (promoted in *atbzip62* and inhibited in *atgsnor1-3*).

### 2.2. In Silico Prediction of Transcription Factor Binding Sites Identified bZIPs Cis-Regulatory Elements

Prior to analyzing the transcripts accumulation of the selected branching related genes, we performed an in silico analysis in order to identify potential binding sites of the bZIP62 TF within the promoter region of the target genes. The results revealed the presence *cis-*regulatory elements specific to the *AtbZIP18* and *AtbZIP69*, among others (Table 1). These two bZIP TFs were selected because their expression was recently suggested to be negatively regulated by the *AtbZIP62* TF in response to salt stress [28]. These results would imply that the *AtbZIP62* TF may interact with the *AtbZIP18* and/or *AtbZIP69* in order to regulate the transcription of each of the target branching genes.

### 2.3. AtbZIP62 Differentially Regulated PIN-FORMED and MAX Encoding Genes in Response to Drought Stress

Auxin polar transport in plants involves many efflux carriers (PIN) proteins [29]. Within the cell, PIN proteins are asymmetrically localized and the directional auxin flow is determined by their polarity. In the current research, we studied the expression of the *Arabidopsis PIN1*, *PIN3,* and *PIN7* in response to drought stress. *AtPIN1* is known for playing an active role in the auxin basipetal transport [30], while *AtPIN3* was suggested to function in the lateral redistribution of auxin [31,32] and *AtPIN7* [33] have been suggested to mediate the lateral re-direction (reflux) of auxin back into the PIN1-dependent auxin transport flow [29,34]. The qPCR results indicate that the expression of *AtPIN1* was downregulated (about 0.4-fold change) in Col-0 WT, but a non-significant change was observed in *atbzip62* background (Figure 2A). Meanwhile, the expression of *AtPIN3* and *AtPIN7* was significantly upregulated (3.7 and 2.7-fold change, respectively) in *atbzip62* knockout plants, while in Col-0 WT a non-significant change was recorded (Figure 2B,C). Another important gene regulating auxin polar transport is *AtAXR1*, the *Arabidopsis* auxin-resistance gene. Here, the expression of *AtAXR1* was significantly downregulated (0.3-fold change) by drought stress in Col-0 WT, but an opposite pattern was recorded in *atbzip62* (upregulated by 4.9-fold change) (Figure 2D). Thus, *AtbZIP62* TF, earlier suggested to positively regulate the adatpive response mechanism towards drought tolerance [27], is believed to negatively control the expression of PIN-FORMED encoding genes in response to drought stress.

In the previous paragraphs, *atbzip62* is shown to have increased branching phenotype under normal growth conditions. With regard to the observed phenotype, we were interested to see how the strigolactone pathway genes known for being involved in the control of bud outgrowth in *Arabidopsis* would be regulated in the *atbzip62* mutant (showing more branches) in response to drought stress. Here, the expression pattern of *AtMAX1* did not change significantly in Col-0, but a significant downregulation (0.13-fold change) was recorded in *atbzip62* (Figure 2E). It was interesting to see that the expression of *AtMAX2* and that of *AtMAX3* genes were significantly upregulated (1.9 and 2,288-fold change, respectively) in Col-0 WT, while being downregulated (0.4 and 0.05-fold change, respectively) in the *atbzip62* knockout plants (Figure 2F,G). Additionally, *AtMAX4* gene was significantly upregulated (19.5-fold change) in *atbzip62,* and upregulated by about 3.2-fold change in Col-0 (Figure 2H).

### 2.4. Drought Stress Differentially Expressed Gibberellic Acid Biosynthetic Genes in Col-0 and atbzip62

The growth-related gibberellins biosynthetic pathway genes, *AtGA2ox1* (GA_2_-oxidase), *AtGA20ox1*, and *AtGA20ox2* control key steps of GAs synthesis in plants. Huang, et al. [35] showed that overexpression of *GA20ox* did not affect the gibberellic acid (GA_4_), which is said to be one of the major bioactive GAs in *Arabidopsis* with GA_1_, GA_9,_ and GA_20_ [36]. The authors suggested that other GAs than GA_4_ would be responsible for the growth phenotype observed in the GA20ox OE plants. Our data in the panel I of the Figure 2 show that *AtGA2ox1* was upregulated by about 1.7-fold change in Col-0 WT, while being downregulated (0.11-fold change) in *atbzip62*.

Because the *atbzip62* showed an increased plant height under normal conditions compared to the Col-0 WT, we were interested to see, in addition to the *AtGA2ox1*, the transcriptional regulation of two other important GA biosynthetic pathway genes, *AtGA20ox1* and *AtGA20ox2* [37] under drought stress conditions. The activity of AtGA20ox has been earlier reported to be regulated by environmental stimuli [38,39,40]. Our data show that the expression of *AtGA20ox1* was downregulated by drought stress in both the Col-0 WT and the *atbzip62* (0.5 and 0.2-fold change, respectively) (Figure 2J), contrasting with the recorded expression of *AtGA20ox2* in both genotypes (significantly upregulated by 1.5-fold change in WT and a non-significant change was recorded in *atbzip62*) (Figure 2K). Further investigations revealed that *AtRGA1* (a member of the GRAS (GIBBERELLIN-INSENSITIVE (RGA), REPRESSOR of ga1-3 (RGA) and SCARECROW (SCR)) transcription factor familiy protein and the VHIID domain/Deletion of five amino acids (VHIID/DELLA) regulatory family [41], known as a repressor of GA signaling, which inhibits the proliferation and expansion of cell-mediated plant growth [42], was induced (1.6-fold change) by drought stress in *atbzip62*, but downregulated (0.7-fold change) in Col-0 WT (Figure 2L).

### 2.5. Drought Stress Differentially Regulated AtIPT5 and AtIPT7 in atbzip62

We expressed three cytokinin biosynthesis pathway genes in Col-0 WT and *atbzip62* under drought stress to explore the possibility for the *AtbZIP62* TF to mediated the transcriptional regulation of isopenteniltransferase (IPT) genes. It is known that the IPT protein controls the rate-limiting step of cytokinin biosynthesis [43,44]. Our data show, on the one hand, that the expression of *AtIPT5* was suppressed at basal level in *atbzip62*, and remained unchanged under drought stress compared to WT showing higher transcript abundance in well-watered plants and suppressed expression in response to drought stress (Figure 2M). On the other hand, *AtIPT7* exhibited an opposite expression pattern under the same conditions. Col-0 plants exposed to drought stress significantly upregulated *AtIPT7* (6.6-fold change)*;* however, when expressed in the *atbzip62*, the transcripts accumulation of *AtITP7* dropped significantly (2.9-fold change) under the same conditions (Figure 2N). 

### 2.6. AtbZIP18 and AtbZIP69 Are Differentially Regulated betwen Col-0 and atbzip62

The bZIP transcription factors are said to either operate alone or in complex with other bZIP TFs to regulate the expression of their target genes. In a recent study, the *AtbZIP18* and *AtbZIP69* were proposed to be negatively regulated by *AtbZIP62* in response to salt stress [28]. Similarly, our data show that the expression of *AtbZIP18* and *AtbZIP69* were upregulated in the *atzip62*, while showing a downregulation pattern in Col-0 wild type (Figure 2O,P).

## 3. Discussion

### 3.1. The AtbZIP62 TF Differentially Regulates the Expression of the Arabidopsis PIN-FORMED Protein, MAX, IPT and GA-Biosynthetic Encoding Genes in Response to Drought Stress

Shoot branching is controlled at different levels by a complex hormonal signaling network, which moves throughout the plant. Auxin efflux carrier PINs play an important role in auxin transport and redistribution to the plant’s organs. PIN1 is the major auxin efflux carrier with a high affinity for auxin polar transport in plants, generating a unidirectional flow of auxin basipetal [34]. Hence, owing to the fact that *AtPIN1* expression was significantly suppressed in the *atbzip62* knockout plants, at both basal level and under drought stress conditions, our data suggest that auxin polar transport through the plant stem to control axillary bud outgrowth might be restricted in *atbzip62* plants. In addition, the recorded transcripts accumulation of *AtPIN3* and *AtPIN7* suggest a possible co-expression in response to drought stress (Figure 2B,C). In *Arabidopsis*, *AtAXR1* was earlier reported to confer auxin resistance to mutant plants. Moreover, *axr1* loss of function mutant has been shown to have a reduced sensitivity to auxin [45]. Therefore, the recorded upregulation of *AtAXR1* in *atbzip62* knockout plants under drought stress would imply that the *AtbZIP62* negatively regulates the expression of the *AtAXR1*, which would result in auxin sensitivity. The expression pattern of *AtAXR1* would also suggest that the increased shoot branching phenotype observed in *atbzip62* would be independent of *AtAXR1*.

From another perspective, a study conducted by Bennett, et al. [46] reported that strigolactone biosynthesis pathway genes, *MAX,* control shoot branching through the regulation of auxin polar transport-dependent to the PIN1 activity, but independent to the activity of AXR1. Similarly, Ferguson and Beveridge [47] studied the roles of major hormones in the regulation of shoot branching, and supported the assumption that strigolactone (SL) and cytokinin (CK) have an antagonistic effect on shoot branching (SL inhibits axillary bud outgrowth, while CK has an opposite effect). 

In a converse approach, a study aiming at characterizing *max1* to *max4* mutants revealed that all *MAX* genes act in the same pathway, with no redundancy in their activity [48]. Generally, *AtMAX1, AtMAX3*, and *AtMAX4* are involved in the synthesis of strigolactone (SL), whereas *AtMAX2* is more active in the signal perception. *AtMAX1* encodes P450, a cytochrome family member, which acts downstream of *AtMAX3* and *4* to yield a carotenoids-derived branching inhibiting hormone. The high nitric oxide (NO) producing mutant *atgsnor1-3* is also known for its increasing branching phenotype. In our CySNO (*S-*nitroso L-cysteine) transcriptome [49], *AtMAX1* was shown to be about 1.94-fold downregulated by NO among other differentially expressed genes (DEGs). Thus, the recorded downregulation of *AtMAX1, AtMAX2,* and *AtMAX3* expressions in the *atbzip62* (Figure 2E–G) would imply that the transcripts accumulation of the three MAX genes could be positively governed by the *AtbZIP62* TF. Moreover, the exponential increase in the transcripts of *AtMAX3* in Col-0 WT by drought stress proposes *AtMAX3* as an emerging candidate gene, which may play a leading role in the adaptive response mechanism towards drought tolerance under the regulatory influence of *AtbZIP62* TF. In the same way, the expression pattern of *AtMAX4* would indicate that all MAX may not co-express under drought stress.

In addition, it has been evidenced that the control of shoot branching is driven by the combinational action of diverse phytohormones, indicating a highly interactive and balanced hormonal signaling cascade [47]. The cytokinin biosynthetic genes, IPTs, were shown to interact with other plant hormones to regulate axillary bud outgrowth. [50]. However, under drought stress, our data shown in panels M and N of Figure 2 suggest a positive regulation of *AtIPT5* and *AtIPT7* by *AtbZIP62* TF, which implies that CK biosynthesis or signaling pathway would be activated under drought stress conditions.

It is well established that plants favor growth and development under normal conditions. Here, we reported an increased plant height phenotype of *atbzip62* plants compared to WT, contrasting with that of the *atgsnor1-3* under normal growth conditions (Figure 1I). However, in response to an environmental stimulus, plants reallocate their resources by activating defense genes, and hormonal signaling as part of the adaptive response towards stress tolerance [51]. In a recent study, the rice RF2a TF (*OsbZIP75*) and RF2b (*OsbZIP30*) were suggested to control GA activity, and consequently plant height [25]. In the same way, Liao, et al. [52] supported that the soybean bZIP TFs *GmbZIP44*, *GmbZIP62*, and *GmbZIP78* negatively regulated abscisic acid (ABA) signaling in transgenic *Arabidopsis* under salt stress. In contrast, the recorded downregulation of *AGA2ox1, AtGA20ox1,* and *AtGA20ox2* in *atbzip62*, which matched the enhanced transcripts abundance of *AtRGA1*, a member of the GRAS (GAI, RGA, SCR) transcription factor family, encoding a DELLA protein [41], suggests their positive regulation by *AtbZIP62* TF under drought stress. In plants, DELLA proteins are growth repressors and modulate all aspects of gibberellic acid responses [53]. This is consistent with the report by Willige, et al. [54], which indicated that a reduction in auxin transport was observed in the inflorescence of *Arabidopsis* mutants deficient in GA biosynthesis concomitant with a decrease in the *PIN* transcripts abundance, which facilitates auxin efflux in GA-deficient plants.

The results of the transcription factors binding sites prediction did not detect any specific *cis-*regulatory element of the *AtbZIP62* TF in the promoter regions of the target genes. Rather, we detected *cis-*regulatory element of two other bZIP TF (*AtbZIP18* and *AtbZIP69*) among others, which have been reported to be negatively regulated by the *AtbZIP62* TF in response to salinity stress [28]. Similarly, data shown in the panels O and P of Figure 2 revealed that the expression of *AtbZIP18* and *AtbZIP69* TFs was significantly upregulated by drought stress in the *atbzip62* knockout plants, but downregulated in the Col-0 WT. Therefore, the presence of the potential binding sites for the *AtbZIP18* and *AtbZIP69* in the promoter of *AtPIN1*, *3*, and *7*, *AtMAX2, 3*, and *4*, *AtAXR1* and *AtIPT5* genes, coupled with the proposed transcriptional interplay between *AtbZIP62* and *AtbZIP18* and *AtbZIP69*, associated with the differential expression patterns of the target branching genes between the *atbzip62* mutant and the Col-0 WT (Figure 2O,P) suggest that *AtbZIP62* TF may require *AtbZIP18* and/or *AtbZIP69* TFs to regulate the expression of more axillary branching (MAX, strigolactone biosynthetic pathway), PIN-FORMED protein (auxin carriers), gibberellic acid biosynthetic pathway genes (GAs), and isopentenyl transferase (IPT, cytokinin biosynthetic pathway) genes in response to drought stress, which would be mediated by the *AtbZIP18* and *AtbZIP69* TF.

### 3.2. Proposed Signaling Model of AbZIP62 TF and Homonal Biosynthesic Genes under Drought Stress

Drought stress response mechanisms are complex and involve a very diverse signaling network and metabolic components, which include phytohormones signaling, activation of abiotic stress-responsive transcription factors and genes, stomata regulation, activation of ROS and reactive nitrogen species (RNS) (including nitric oxide, NO) signal pathways. All these metabolic components interact in a very coordinated and sophisticated manner to ensure that the plant uses efficiently the energy produced to maintain its fitness and survival. In Figure 3, this study proposed a signaling model involving the *AtbZIP62* TF and other phytohormones biosynthetic pathway genes in response to drought stress of which the transcripts accumulation was measured in the current work. In our CySNO transcriptome [49], an *Arabidopsis* bZIP transcription factor (*AtbZIP53*, AT3G62420) encoding gene was shown to be upregulated by 2.61-fold. Similarly, our recent study showed that a significant reduction in SNO (*S-*nitrosoglutathione) and abscisic acid (ABA) contents was observed in the *atbzip62* knockout plants compared to Col-0 WT [27]. Furthermore, NO was reported to interact with auxin to regulate roots growth in rice [55]. In the same way, NO was suggested to mediate the cytokinin functions in cell proliferation and meristem maintenance in *Arabidopsis* [56], while strigolactone was suggested to interact with NO in regulating root system architecture of *Arabidopsis* [57]. Moreover, crosstalk between NO and phytohormones during plant development has been widely discussed [58]. Therefore, the signaling model proposed in the present study tends to explain how *AtbZIP62* TF would regulate the MAX, PIN-FORMED, GA, and IPT encoding genes under drought stress conditions. This model was designed based on the recorded transcriptional regulation patterns of the genes studied, monitored by qPCR in the *atbzip62* knockout plants, coupled with previously reported evidence. Generally, plant hormones are part of the adaptive response mechanisms towards stress tolerance and they interact either in synergy or antagonize each other. Abscisic acid (ABA) has been extensively studied and has been shown to be one of the most responsive phytohormones under abiotic stress in plants. Our findings open new paths towards understanding the role of other hormonal signaling pathways genes in response to stress (abiotic), in addition to their roles in the regulation of plants growth and development.

## 4. Materials and Methods

### 4.1. Plant Materials and Growth Conditions 

The seeds of the wild type (WT) *Arabidopsis* Col-0 and the *atbzip62* (AT1G19490: SALK_053908C) knockout line derived from it were obtained from the Arabidopsis Biological Resource Center (ABRC) (https://abrc.osu.edu/ (accessed on 31 January 2021)). In addition, the *atgsnor1-3* mutant lacking the *S-*nitrosoglutathione (GSNO) reductase 1 (*GSNOR1*), known to regulate the cellular *S-*nitrosothiols (SNO) levels, was included for its stunt, and high branching phenotype was identified from the GABI-Kats T-DNA insertion collection [59,60]. Plants were grown on a peat moss soil mixture at 22°C with 16 h light and 8 h dark cycles. The *atbzip62* plants were previously genotyped to identify homozygous transfer DNA (T-DNA) insertion lines by polymerase chain reaction (PCR) for further experiments. The T-DNA insertion lines were confirmed as described earlier [27]. The list of primers and their corresponding forward and reverse sequences is given in Table 2.

### 4.2. In Silico Transcription Factor Binding Site Prediction

With regards to the transcripts accumulation levels of the target branching related genes, we performed an *in silico* analysis in order to explore the possibility for the *AtbZIP62* TF to be involved in their transcriptional initiation or expression, using bioinformatics approach. To achieve that, we predicted the transcription factor binding sites in the DNA sequence of each of the selected hormonal signaling pathway genes included in the study using the Binding site prediction tool available at http://plantregmap.gao-lab.org (accessed on 31 January 2021).

### 4.3. Drought Stress Induction by Water Withholding

Plants were subjected to drought stress at rossette stage (4-week-old plants) following the water withholding method as described earlier [61], with slight modifications. Briefly, the moisture content of the soil was routinely monitored by measuring the weight of each of the 50-well trays in triplicate to evaluate the water loss. The soil moisture percentage (about 30%) was calculated as the percentage of the actual weight loss relative to the initial weight of the saturated soil considered as having 100% moisture content. Leaf samples for gene expression analysis were collected as soon as the loss of turgidity and wilting of leaves were apparent (9 days after water withholding).

### 4.4. Total RNA Isolation and Gene Expression Analysis by qPCR

Total RNA was isolated from leaf samples collected a rosette stage, using the TRI-Solution^TM^ Reagent (Cat. No: TS200-001, Virginia Tech Biotechnology, Lot: 337871401001) following the manufacturer’s instructions. Thereafter, the complementary DNA (cDNA) was synthesized as previously described [62]. The cDNA was then used as a template in qPCR to study the transcripts accumulation of selected genes (Table 2) using SYBR green (BioFact, Daejeon, Korea) in a real-time PCR machine (Eco™ Illumina, San Diego, CA, USA). The relative expression of each gene was normalized to that of the *Arabidopsis* actin.

## 5. Conclusions

For the last two decades, the focus of plant biosciences-related research has been marked by an increasing interest in the use of a genome-wide approach while studying the function of genes in response to adverse environmental conditions, rather than a single gene–gene approach. The output from these years of research helped elucidate factors involved in the regulation of gene expression, including the cross-talk mechanism.

This study investigated the transcriptional regulation of key branching related genes in response to drought stress, including *AtPIN1, 3,* and *7*, *AtMAX1–4*, *AtXR1*, *AtIPT5* and 7, *AtGA2ox1*, *AtGA20ox1* and *2*, and *AtRGA1*. The results showed that all the branching-related genes were differentially regulated between Col-0 WT and the *atbzip62* mutant in response to drought stress. In addition, with regard to the transcripts accumulation patterns of the two bZIP TFs, *AtbZIP18* and *AbZIPT69,* in the *atbzip62* background, and their proposed transcriptional interplay, coupled with the detected potential binding sites specific to the *AtbZIP18* and *AtbZIP69* in the target genes, all the results suggest that the *AtbZIP62* TF may require *AtbZIP18* and/or *AbZIP69* to regulate the expression of the analyzed branching related genes under drought stress conditions. Future studies may include the use of loss of function mutant lines of selected hormonal signaling pathways genes, other than abscisic acid (*atmax*, *atpin*, or *atipt*) reported in the current study to elucidate their roles in drought stress response.

## Figures and Tables

**Figure 1 genes-12-00298-f001:**
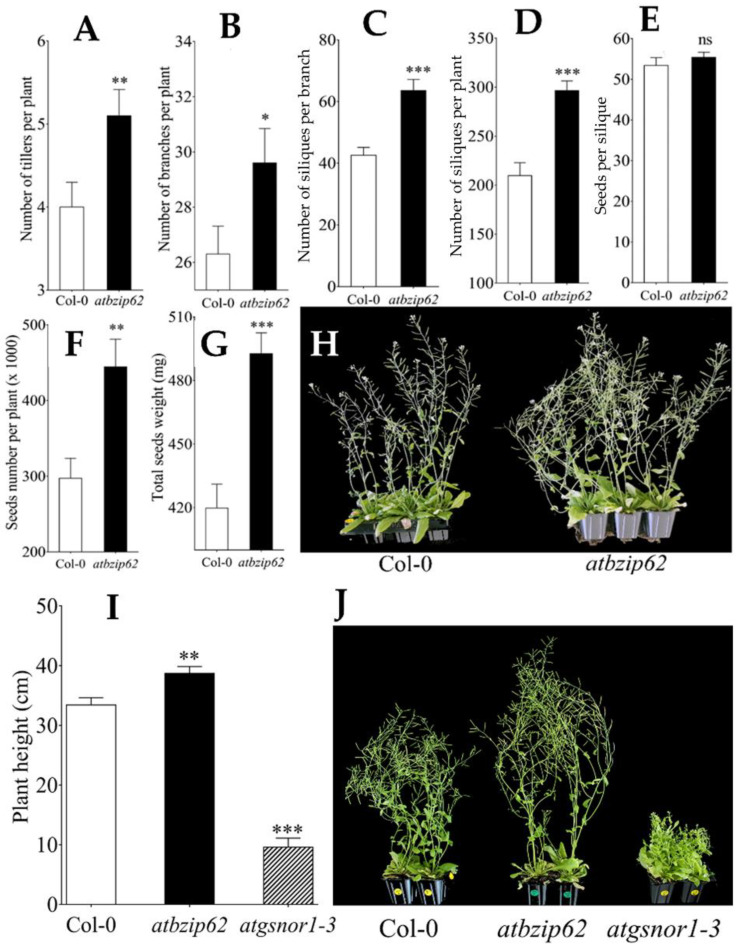
Morphology of *atbzip62* plants grown under normal conditions. (**A**) Number of tillers per plant, (**B**) number of branches per plant, (**C**) number of siliques per branch, (**D**) number of siliques per plant, (**E**) number of number seeds per silique, (**F**) number of seeds per plant, (**G**) seeds weight per plant, (**H**) phenotype of Arabidopsis *atbzip62* showing an increased number of tillers compared to Col-0 wild type, (**I**) plant height, and (**J**) phenotype of *atbzip62* plants showing an increased in plant height compared to Col-0 and *atgsnor1-3*. *** *p* < 0.001, ** *p* < 0.01, * *p* < 0.05, ns non-significant.

**Figure 2 genes-12-00298-f002:**
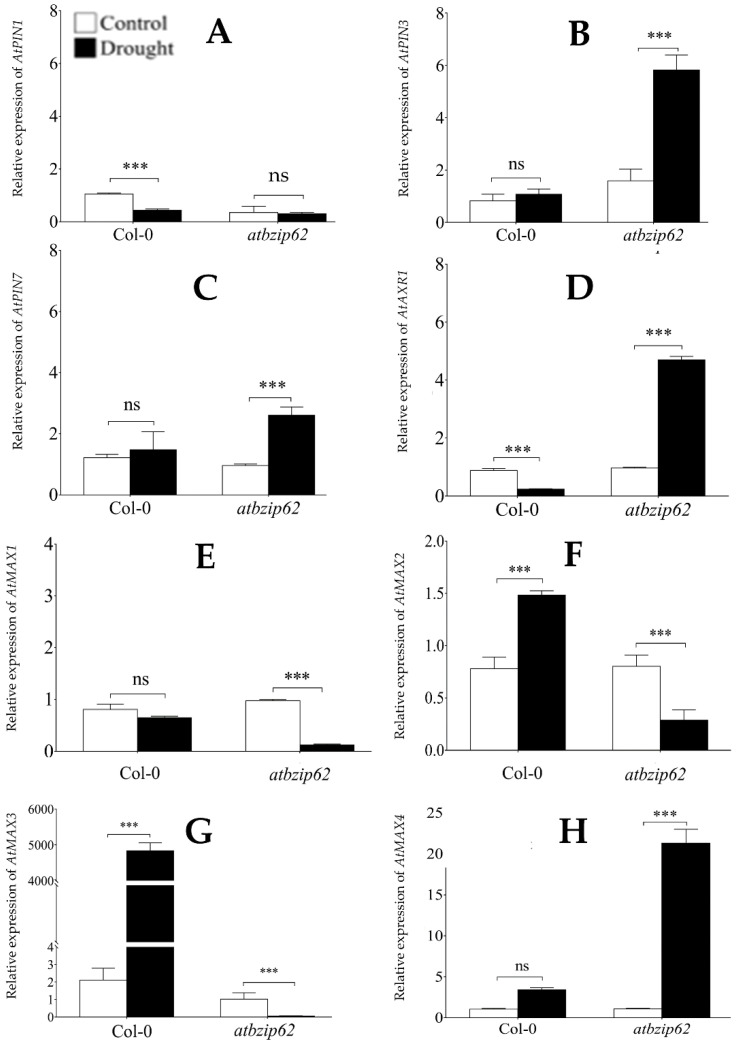
Transcripts accumulation of hormonal responsive genes under drought stress conditions. (**A**) AtPIN1. (**B**) *AtPIN3*, (**C**) *AtPIN7*, (**D**) *AtAXR1*, (**E**) *AtMAX1*, (**F**) *AtMAX2*, (**G**) *AtMAX3*, (**H**) *AtMAX4*, (**I**) *AtGA2ox1*, (**J**) *AtGA20ox1*, (**K**) *AtGA20ox2*, (**L**) *AtRGA1*, (**M**) *AtIPT5*, (**N**) *AtIPT7*, (**O**) *AtbZIP18*, and (**P**) *AtbZIP69* in *Arabidopsis* Col-0 wild type and *atbzip62* knockout plants exposed to drought stress. Bars are mean values ± SE. White bars are controls (routinely watered) plants and black bars are drought-treated plants in triplicate. *** *p* < 0.001, ** *p* < 0.01, * *p* < 0.05, ns non-significant.

**Figure 3 genes-12-00298-f003:**
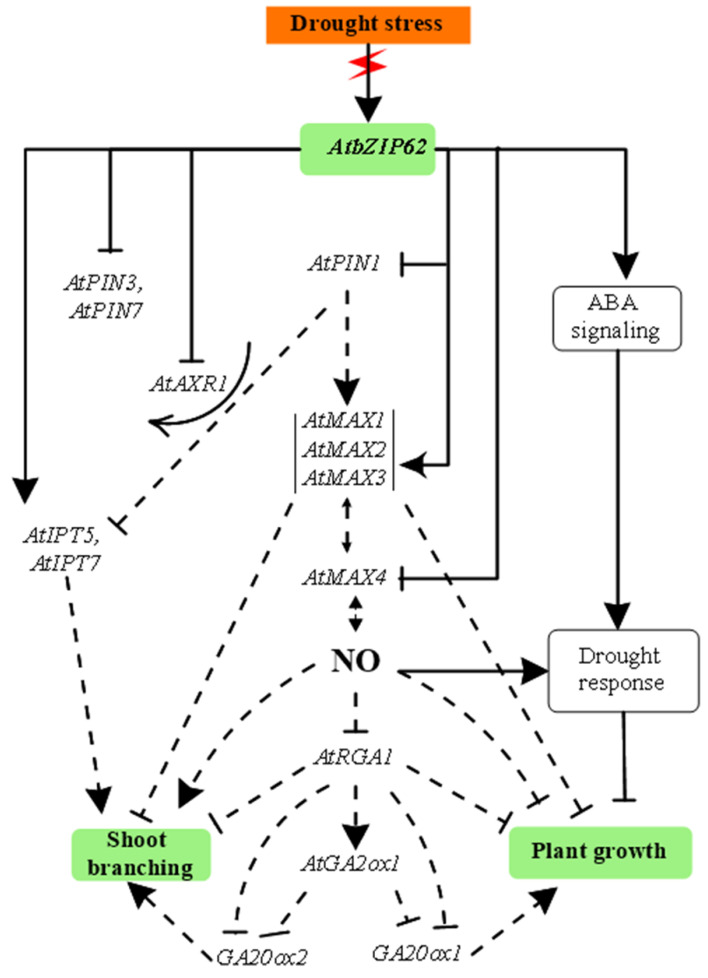
Signaling model involving *AtbZIP62* transcription factor (TF) and more axillary branching (MAX), PIN-FORMED, gibberellic acid (GA), and isopentenyl transferase (IPT) encoded genes under drought stress. Upon drought stress induction, a variety of signals transduction and hormonal pathways are activated, followed by the induction of an array of drought-responsive genes, including transcription factors, such as *AtbZIP62* which interact with other proteins or DNA to regulate the expression of stress inducible genes. As part of the drought response mechanism, plants activate ABA signaling components, which interact with other signaling pathways and specific genes in order to provide an appropriate response towards stress tolerance. Consequently, the regular plant growth-related metabolism is reduced. In the proposed signaling model, *AtbZIP62* TF is shown to differentially regulate the transcripts accumulation of *AtMAX1*, *AtMAX2*, *AtMAX3*, *AtMAX4* (SL), *AtPIN1*, *AtPIN3*, *AtPIN7* (auxin carriers), *AtGA2ox1*, *AtGA20ox1*, *AtGA20ox1*, and *AtRGA1*, *AtIPT5,* and *AtIPT7* (Cytokinin) genes. Arrows with continuous lines indicate positive regulation (of gene expression or induction of plant growth/shoot branching). Continuous lines with a perpendicular bar or an arrow suggest a negative regulation/inhibition or positive regulation/induction (of gene expression or plant growth/shoot branching) by our studies (current and previous). Dotted lines with a perpendicular bar or an arrow suggest a negative regulation/inhibition or positive regulation/induction (of gene expression or plant growth/shoot branching) by previous evidence by other research groups. NO, nitric oxide.

**Table 1 genes-12-00298-t001:** Transcription factor binding sites prediction.

Gene ID	Gene Name	Target Genes	Position	Strand	*p*-Value	q-Value	Matched Sequence
*	*	*AtMAX1*	*	*	*	*	*
*	*	AT2G26170	*	*	*	*	*
***AtMAX2***							
AT2G40620	*AtbZIP18*	AT2G42620	1723–1733	-	2.14 × 10^−5^	0.046	CTCGGCTGGCC
AT2G40620	*AtbZIP18*	AT2G42620	923–933	+	7.04 × 10^−6^	0.0303	TTCAGCTGTCA
AT1G06070	*AtbZIP69*	AT2G42620	923–933	+	1.41 × 10^−5^	0.0583	TTCAGCTGTCA
***AtMAX3***							
AT2G40620	*At* *bZIP18*	AT2G44990	1704–1714	-	6.59 × 10^−5^	0.295	AATAGCTGTCG
***AtMAX4***							
AT2G40620	*AtbZIP18*	AT4G32810	2535–2545	-	2.48 × 10^−5^	0.158	CACGGCTGTCT
AT2G40620	*AtbZIP18*	AT4G32810	285–295	+	8.64 × 10^−5^	0.276	CAGAGCTGTAA
AT1G06070	*AtbZIP69*	AT4G32810	2535–2545	-	6.02 × 10^−5^	0.369	CACGGCTGTCT
***AtPIN1***							
AT1G06070	*AtbZIP69*	AT1G73590	1935–1945	-	2.08 × 10^−5^	0.138	AAGAGCTGGCA
***AtPIN3***							
AT2G40620	*AtbZIP18*	AT1G70940	2256–2266	-	1.97 × 10^−5^	0.131	CCCACCTGTCG
AT1G06070	*AtbZIP69*	AT1G70940	2256–2266	-	5.15 × 10^−5^	0.336	CCCACCTGTCG
***AtPIN7***							
AT2G40620	*AtbZIP18*	AT1G23080	49–59	+	9.48 × 10^−5^	0.628	AAAAGCTGTAA
***AtAXR1***							
AT2G40620	*AtbZIP18*	AT1G05180	1160–1170	+	4.52 × 10^−5^	0.338	ACCACCTGTCT
AT1G06070	*AtbZIP69*	AT1G05180	1035–1045	+	4.37 × 10^−5^	0.254	TGCAGCTGGTG
AT1G06070	*AtbZIP69*	AT1G05180	1160–1170	+	6.95 × 10^−5^	0.254	ACCACCTGTCT
***AtIPT5***							
AT1G06070	*AtbZIP69*	AT5G19040	1294–1304	-	8.78 × 10^−5^	0.292	GGCGGCTGGAA
*	*	*AtGA2ox1*	*	*	*	*	*
*	*	*AtGA20ox1*	*	*	*	*	*
*	*	*AtGA20ox2*	*	*	*	*	*
*	*	*AtRGA1*	*	*	*	*	*

(*) indicates that the binding sites for the *AtbZIP18* and *AtbZIP69* were not detected.

**Table 2 genes-12-00298-t002:** List of primers for expression of target genes used in the study.

Gene Name/Genotype	Locus/ SALK	Forward Primer (5′->3′)	Reverse Primer (5′->3′)	Gene Name
*atbzip62*	SALK_053908C	TGGCACTTTTAACTTTGTGCC	TACGTTTCCATCGAGTGAACC	Arabidopsis bzip62 loss of function mutant
*Drought responsive gene in Arabidopsis*				
*AtbZIP62*	*AT1G19490*	*CATCGAGTTGTTGCTCGTCG*	*AAATCCGCCAATGCTTCTGC*	Basic-leucine zipper transcription factor encoding gene 62
*Genes involved in strigolactone biosynthesis pathway and controlling shoot branching (bud outgrowth)*				
*AtMAX1*	AT2G26170	TGGTCACTTGCCCTTGATGG	GGTTGCCTCCCCATCTGAAA	More axillary branching 1 gene
*AtMAX2*	AT2G42620	CCGAGCCAGAGTTTGGGTTA	GTGCGAAACCGATTGTGTCC	More axillary branching 2 gene
*AtMAX3*	AT2G44990	CGTTGGTGAGCCCATGTTTG	TCCACCGAAACCGCATACTC	More axillary branching 3 gene
*AtMAX4*	AT4G32810	TATCGGGTCGTGAGGATGGA	GCAAACGAATGGACCCAACC	More axillary branching 4 gene
Genes involved in Auxin polar transport (efflux carrier) and controlling shoot branching				
*AtPIN1*	AT1G73590	ACGACAACCAGTACGTGGAG	TATGTTGTTCCCACCGTCCG	PIN-FORMED (*PIN*) *protein encoding gene1*
*AtPIN3*	AT1G70940	TGGCCATGATCCTCGCTTAC	CGAAGATGGCGACAAAACGG	PIN-FORMED (*PIN*) *protein encoding gene 3*
*AtPIN7*	AT1G23080	AGCCATGATCCTCGCTTACG	AGAGGGACGGCGAAAATAGC	PIN-FORMED (*PIN*) *protein encoding gene 7*
Cytokinin biosynthetic pathway genes				
*AtIPT5*	AT5G19040	CGACGGAGGTTTTTCTCCGA	GAACTTTTCGACGGCGAGTG	Isopentenyl transferase encoding gene 5
*AtIPT7*	AT3G23630	GACGCCACTGAGGTGTTCTT	CGACGATTCTCTCGCTTGGT	Isopentenyl transferase encoding gene 7
Genes involved in Gibberellic acid biosynthesis pathway				
*AtGA2ox1*	AT1G78440	CTCGTTGCCCAAGTCAGAGA	TACTCAACCCAACCCACGTC	Gibberellic acid 2 oxidase 1
*AtGA20ox1*	AT4G25420	GTGAGAGTGTTGGCTACGCA	CTCATGTCGTCGCAAAACCG	Gibberellic acid 20 oxidase 1
*AtGA20ox2*	AT5G51810	TGGCCAGACGAAGAGAAACC	TTGACGACGAGGAAGAAGCC	Gibberellic acid 20 oxidase 2
*AtAXR1*	AT1G05180	CGGACAGATTTGCTGCCAAC	ATCTGGGAGTACTGAGCCGT	Arabidopsis Auxin repressor 1
*AtRGA1*	AT2G01570	TTGTCCAACCACGGGACTTC	AGCTCGTCGTCCATGTTACC	Arabidopsis Repressor of GA 1
Arabidopsis housekeeping gene				
*AtACT2*	AT3G18780	AGGTTCTGTTCCAGCCATC	TTAGAAGCATTTCCTGTGAAC	Arabidopsis Actin coding gene 2

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
