# Peer review of "Insights into the Transcriptional Regulation of Branching Hormonal Signaling Pathways Genes under Drought Stress in Arabidopsis"

_genes, 2021, doi:10.3390/genes12020298_

Round 1

Reviewer 1 Report

The manuscript by Rolly et al studied bZIP62, a transcriptional factor was found to regulate drought stress response. The authors first predicted potential binding sites of bZIP62. Among the potential bZIP62 binding genes, branching related genes were significantly represented. Consistent with their in silico prediction, the authors found that bzip62 mutant showed elevated number of branches, pods, seeds, etc. Then, the authors measured the expression of these genes under drought stress. In general, I feel the design of this study is good. However, I have two major concerns about this manuscript: 1) data presentation and writing need to be improved; 2) overstatement in Figure 3.

Section 2.1

The authors listed the potential binding genes of bZIP62 at the beginning. But suddenly, they talking about bZIP18 and bZIP69. I understand that the authors try to say that their prediction is good, which is supported by previous experimentally validated data, bZIP18 and bZIP69.

Section 2.2

The authors suddenly describe the branching phenotype of bzip62 mutant. I guess the reason the authors doing this experiment is based on the gene list in Table 1. In order to make the readers understand this story easily, I strongly suggest the authors pay more attention to improve the logic of this story.

Fig 1J, why show the phenotype of atgsnor1-3? Very confusing!

Figure 3, I suggest the authors delete Figures, because most of them are lack of evidence. If the authors want to propose a model, please provide solid evidence.

Author Response

Insights into the Transcriptional Regulation of Branching Hormonal Signaling Pathways Genes under Drought Stress in Arabidopsis Manuscript ID: genes-1106328 Point by point reply to comments of reviewers We are thankful to the editorial team and anonymous reviewers for their time given to this manuscript. We appreciate their comments, and are happy to share that most of the comments are addressed and have substantially improved the quality of the manuscript. We would like to specify that all changes in the manuscript were highlighted green. No track change was applied to the manuscript. We hope that the manuscript in the present form will be suitable for publication in the journal. Reviewer 1 The manuscript by Rolly et al studied bZIP62, a transcriptional factor was found to regulate drought stress response. The authors first predicted potential binding sites of bZIP62. Among the potential bZIP62 binding genes, branching related genes were significantly represented. Consistent with their in silico prediction, the authors found that bzip62 mutant showed elevated number of branches, pods, seeds, etc. Then, the authors measured the expression of these genes under drought stress. In general, I feel the design of this study is good. However, I have two major concerns about this manuscript: 1) data presentation and writing need to be improved; 2) overstatement in Figure 3. The authors would like to thank the reviewer for his positive and constructive comments. We would like to share that almost all the comments were addressed, and the quality of the manuscript has been substantially improved as suggested. Section 2.1 The authors listed the potential binding genes of bZIP62 at the beginning. But suddenly, they talking about bZIP18 and bZIP69. I understand that the authors try to say that their prediction is good, which is supported by previous experimentally validated data, bZIP18 and bZIP69. Now 2.2 (line 131) The authors would like to apologize for the inconvenience. Therefore, we have modified the paragraph under the section 2.1 (see lines 131-140) Section 2.2 The authors suddenly describe the branching phenotype of bzip62 mutant. I guess the reason the authors doing this experiment is based on the gene list in Table 1. In order to make the readers understand this story easily, I strongly suggest the authors pay more attention to improve the logic of this story. 2.1. The authors are thankful to the reviewer, and agreed with the suggestion. Therefore, the previous section 2.2 is now moved upward and changed to 2.1 (lines 107-122). Fig 1J, why show the phenotype of atgsnor1-3? Very confusing! The authors apologize for the inconvenience. We would like to indicate that the reason for comparing the growth phenotypes of atgnor1-3 with that of the well characterized NO deficient mutant known for its roles in the positive regulation of shoot branching. The atgsnor1-3 mutant is a high nitric oxide producing mutant compared to the Col-0 wild type. This mutant exhibit an increased branching phenotype, but show a contrasting growth pattern with the atbzip62. In a recent study, atbzip62 was shown to produce less nitric oxide (NO) (endogenous SNO level) compared to the wild type and atgsnor1-3. Unfortunately, we have been urged not use the data related to molecular, physiological or biochemical responses of atgsnor1-3 mutant under abiotic stress conditions, including drought stress, to avoid any form conflict of interest with the research group that generated the atgsnor1-3 mutant, from which we have received the seeds of the atgsnor1-3, except for biotic or nitro-oxidative stress related experiments. This research group is currently characterizing the atgsnor1-3 (lacking the OsGNOR1 regulating to endogenous NO level) under abiotic stress conditions (salinity, drought, etc.). Reporting data related mechanism of NO in response to abiotic response mechanism using the atgsnor1-3 (a high NO producing mutant), may dilute their story. So, we were left with no other choice than to use the only phonotype of the atgsnor1-3 showing a stunt phenotype to compare with the atbzip62. We have tried to give an explanation in lines 118-122. Figure 3, I suggest the authors delete Figures, because most of them are lack of evidence. If the authors want to propose a model, please provide solid evidence. The authors appreciate the concern raised by the reviewer. We would like to indicate that the proposed signaling model is consisted, on the one hand of current available literature and general knowledge regarding the regulation of shoot branching in plants, and represented by the dotted (discontinuous) lines. The roles of each phytohormone in the regulation of shoot branching has been largely investigated and reported. We have tried to put together this information and discuss some of the reported evidences in the discussion section. On the other hand, based on the recorded transcripts accumulation patterns of the target branching genes between the wild type (Col-0) and the mutant (atbzip62) in response to drought stress, we proposed a point of connection between the previous reports and the present brief report, which opens new paths for more elaborated research to elucidate the roles of each of the studied genes.

Reviewer 2 Report

The authors with this study tried to investigate the transcriptional regulation of hormonal biosynthesis pathway genes. In this context they focused their attention on a specific AtbZIP TF, AtbZIP62, known to be involved in the adaptive response in drought stress conditions.

To do this their analysed:

  • Possible AtZIP62 target’s;
  • Evaluate the expression patterns (in drought condition) of these targets in a knock-out line (Atbzip62).

The manuscript is interesting, however, it still needs some modify.

Abstract:

The explanation of why they selected atbZIP62 as a target gene is not exhaustive (see lines 20 to 23, this sentence could be edited)

Introduction:

Line 48: Please edit “produciton” with “production”;

Line 50: “damge” with “damage”

Lines 57-58: Please add a reference;

Results:

  • I suggest swapping paragraph 2.2 in 2.1. It is preferable to know which are the phenotypical characteristics of this mutant line before to report possible linkages with other TF.
  • Move paragraph 2.1 (“in silico prediction […]”) as the last part of the Result section;
  • Please rename “pods” with “siliques” (pods is unusual terms for Arabidopsis) ;
  • In fig.1-J the authors compared the phenotypes of the atZIP62 with the WT and atsnor1-3. This specific mutant line is not reported in material and method. I know it was reported with a linked ref, nevertheless, could be easier for the reader find this information in this manuscript;
  • Figure 2: there is no caption for the expression pattern of figure O and P. Please add it.

Discussion:

General comment: It is not clear, from my point of view, the role of the cis-regulatory elements, bZIP18 and 69. Line 280-282 seems to explain their role, but no refs are reported.

- Line 239: “NO” specify here the acronyms (not at line 241);

- Line 263: “study”

Material and method:

- Line 347: “rosette”;

- Line 348: “method”

- RNA was isolated from the leaves. What was the stage of these leaves? Please specify it.

Author Response

Insights into the Transcriptional Regulation of Branching Hormonal Signaling Pathways Genes under Drought Stress in Arabidopsis

Manuscript ID: genes-1106328

Point by point reply to comments of reviewers

We are thankful to the editorial team and anonymous reviewers for their time given to this manuscript. We appreciate their comments, and are happy to share that most of the comments are addressed and have substantially improved the quality of the manuscript. We would like to specify that all changes in the manuscript were highlighted green. No track change was applied to the manuscript. We hope that the manuscript in the present form will be suitable for publication in the journal.

Reviewer 2

The authors with this study tried to investigate the transcriptional regulation of hormonal biosynthesis pathway genes. In this context they focused their attention on a specific AtbZIP TF, AtbZIP62, known to be involved in the adaptive response in drought stress conditions.

To do this their analyzed:

·         Possible AtZIP62 target’s;

·         Evaluate the expression patterns (in drought condition) of these targets in a knock-out line (Atbzip62).

The manuscript is interesting, however, it still needs some modify.

We would like to thank the reviewer for his valuable comments and suggest to improve our manuscript. We would like to share that almost all the comments have been addressed, and the manuscript has been improved as suggested.

Abstract:

The explanation of why they selected atbZIP62 as a target gene is not exhaustive (see lines 20 to 23, this sentence could be edited)

We appreciate the comments of the reviewer. We have improved the explanation as suggested (lines 17-22; 26-28)

Introduction:

Line 48: Please edit “produciton” with “production”;

Line 50: “damge” with “damage”

Lines 57-58: Please add a reference;

Line 53, we have corrected as suggested.

Line 55, we have replaced damge with damage as suggested.

Line 63: we have included the references as suggested.

Results:

·         I suggest swapping paragraph 2.2 in 2.1. It is preferable to know which are the phenotypical characteristics of this mutant line before to report possible linkages with other TF.

·         We are thankful to the reviewer for his suggestion. We have no objection in moving the previous section 2.2 to 2.1. The phenotype results are now 2.1 as suggested (lines 107-122)

Move paragraph 2.1 (“in silico prediction […]”) as the last part of the Result section;

·         The authors appreciate the suggestion made by the reviewer. However, would like to indicate that it appears to us that it will make more sense if the in silico results precede the gene expression data, considering that prior to perform the expression analysis, we investigated the transcription binding sites of the bZIP TF in the promoter regions of the target branching related genes.

Please rename “pods” with “siliques” (pods is unusual terms for Arabidopsis) ;

·         We have replaced pods with siliques lines 114 and 115 and in the Figure 1, the caption (lines125-126) as suggested

·         In fig.1-J the authors compared the phenotypes of the atZIP62 with the WT and atsnor1-3. This specific mutant line is not reported in material and method. I know it was reported with a linked ref, nevertheless, could be easier for the reader find this information in this manuscript;

We are thankful to the reviewer for the suggestion. We have included in lines 364-367 the required information of the atgsnor1-3 mutant.

Figure 2: there is no caption for the expression pattern of figure O and P. Please add it.

We would like to apologize for the inconvenience. We have included the caption of the figure panels O and P as suggested (line 220) preceded by the description of the results in lines 205-210.

Discussion:

General comment: It is not clear, from my point of view, the role of the cis-regulatory elements, bZIP18 and 69. Line 280-282 seems to explain their role, but no refs are reported.

We are thankful to the reviewer for the concern raised. We have included a paragraph to create a linkage between the AtbZIP62 the other two bZIP transcription factors AtbZIP18 and AtbZIP69 in lines 293-305.

- Line 239: “NO” specify here the acronyms (not at line 241);

Lin 257: we have specified the acronym “NO” as suggested, and mentioned only NO in line 260.

- Line 263: “study”

We would like to apologize for the inconvenience. We have corrected the spelling as suggested (line 281)

Material and method:

- Line 347: “rosette”;

Line 375, we have corrected the spelling as suggested.

- Line 348: “method”

Line 376, we have corrected the spelling as suggested.

- RNA was isolated from the leaves. What was the stage of these leaves? Please specify it.

Line 384: we have specify that leaf samples were collected at rosette stage

Round 2

Reviewer 1 Report

The authors answered almost all my concerns. I am fine with keeping. Figure 3, but my opinion is that evidence based model is much more better for a research paper. If this is a review paper, I am totally fine.

Reviewer 2 Report

I would like to thank the authors for their effort in improving the manuscript as suggested. The present version is now acceptable for the publication.